https://doi.org/10.1038/s42003-021-02049-6　　**OPEN**
# The protected physiological state of intracellular Salmonella enterica persisters reduces host cell-imposed stress

Marc Schulte [1,2], Katharina Olschewski [1] & Michael Hensel [1,2] ✉

During infectious diseases, small subpopulations of bacterial pathogens enter a non-replicating (NR) state tolerant to antibiotics. After phagocytosis, intracellular Salmonella enterica serovar Typhimurium (STM) forms persisters able to subvert immune defenses of the host. Physiological state and sensing properties of persisters are difficult to analyze, thus poorly understood. Here we deploy fluorescent protein reporters to detect intracellular NR persister cells, and to monitor their stress response on single cell level. We determined metabolic properties of NR STM during infection and demonstrate that NR STM persisters sense their environment and respond to stressors. Since persisters showed a lower stress response compared to replicating (R) STM, which was not consequence of lower metabolic capacity, the persistent state of STM serves as protective niche. Up to 95% of NR STM were metabolically active at beginning of infection, very similar to metabolic capacity of R STM. Sensing and reacting to stress with constant metabolic activity supports STM to create a more permissive environment for recurrent infections. Stress sensing and response of persister may be targeted by new antimicrobial approaches.

[1] Abt. Mikrobiologie, Universität Osnabrück, Osnabrück, Germany. [2] CellNanOs – Center of Cellular Nanoanalytics Osnabrück, Universität Osnabrück, Osnabrück, Germany. ✉email: Michael.Hensel@uni-osnabrueck.de

The global increase of multi-resistant bacteria presents one of the major challenges to human health in the near future. In addition, physicians frequently encounter bacterial infections that are very difficult to treat, and often relapse without the presence of genetic resistance to antibiotics[1–5]. These recurrent infections can only be defeated by several rounds of antibiotic treatment, possibly due to the presence of antibiotic-tolerant persister cells. In the context of recurrent infections, the heterogeneous phenomenon of bacterial antibiotic persistence is becoming increasingly important. Antibiotic persistence describes a phenomenon in which a small part of a bacterial population enters a non-replicating (NR) state that can survive actually lethal concentrations of antibiotics during infection. Persister cells arise from a genetically clonal bacterial population by a transient and reversible phenotype switch, leading to a NR and multidrug-tolerant subpopulation[6,7] reviewed in the ref. [8].

Recent in vitro investigations of antibiotic persistence observed bacteria entering a dormant state when grown in laboratory media[9–11]. However, the physiological state and stress sensing properties of intracellular persistent bacteria are still poorly understood. Many questions regarding the interface of antibiotic persisters with their host remain to be answered, and there is demand for sensitive tools to interrogate the interplay of both organisms. Better understanding of the physiology of persisters will help to device new forms of antimicrobial therapy.

The facultative intracellular pathogen *Salmonella enterica* causes acute and chronic infections[12], and *S. enterica* serovar Typhimurium (STM) serves as a model organism for a facultative intracellular pathogen forming persister cells during systemic infections. It was shown that intracellular persister cells of STM do not show complete dormancy, but consist of subpopulations of metabolically active bacteria, as well as inactive cells showing decreasing responsiveness to external stimuli over the course of infection[13]. Similar NR but metabolically active bacteria have also been observed in macrophages infected with *Mycobacterium tuberculosis*[14]. It was shown that many persisters of STM are formed immediately upon phagocytosis by macrophages. Vacuolar acidification as well as nutritional deprivation was mentioned as one of the main factors leading to macrophage-induced persister formation[15]. More recently, Stapels et al.[16] reported that persisters of STM translocate effector proteins of the *Salmonella* Pathogenicity Island 2(SPI2)-encoded type III secretion system (T3SS), to dampen proinflammatory innate immune response and induce anti-inflammatory macrophage polarization. Such reprogramming of their host cells allows NR STM to survive and might lead to an advantage during infection relapse after termination of antibiosis[16].

Within phagocytic cells, STM encounters harsh environmental conditions and various defense mechanisms including antimicrobial peptides and the respiratory burst[17–19]. For the pathogen it is of crucial importance to sense and react to potentially detrimental factors. Hence, STM has evolved a plethora of defensive virulence mechanisms to withstand antimicrobial effectors and to overcome the clearance by the host reviewed in the ref. [20]. These stress response systems (SRS) are able to sense harmful conditions, as well as perturbations of the bacterial envelope, in periplasm, or in cytoplasm. SRS have to perform efficiently in space and time for successful survival of STM within hazardous host environments. An important protein for the proteolytic degradation of misfolded or damages proteins in the periplasm is the serine protease HtrA, a.k.a. DegP, and functions as a multifunctional protein quality control factor[21]. During the cytosolic stress response, the thioredoxin TrxA and the methionine sulfoxide reductase MsrA play an important role in antioxidant defense reviewed in the refs. [22,23]. TrxA reduces a disulfide bond in an oxidized substrate protein, whereas MsrA is responsible for reduction of oxidized methionine residues[24–26]. Furthermore, the transcriptional regulator DksA is important in repair mechanisms for reactive nitrogen species-mediated damages. When deleted, ΔdksA strains show susceptibility to reactive nitrogen species and attenuated virulence[27,28].

Despite being equipped with an extensive set of defensive and offensive virulence factors, the individual fate of intracellular STM is highly diverse[29]. It was shown that stress response and formation of persister cells are closely related. Among other, main mediators for persister formation are considered to be the SOS response via RecA and LexA, the stringent response via (p)ppGpp, the oxidative stress response via OxyR and SoxSR, and toxin–antitoxin modules reviewed in the refs. [8,30,31]. But what happens after establishment of the persistent status? Is switch from persister state to normal growth a merely stochastic event, or can persisters monitor their environment with the ability to respond to cues? Addressing these questions is challenging due to population heterogeneity of intracellular STM, the low frequency of persisters, and their low metabolic activity.

Sensing of environmental stimuli and especially stressors are of crucial importance for pathogens to maintain their cell integrity. So far, little is known about the stress response of intracellular NR STM. We investigated whether NR STM after entering persistence are still capable in sensing stress factors and in responding by inducing stress responses. If so, is the level of stress response of NR STM similar to replicating (R) STM, or altered? To address these questions, we developed further the recently introduced reporter system[32] to enable analyses of stress response of intracellular STM persister cells. Here, we introduce dual-fluorescence reporters which enable extremely sensitive flow cytometric analyses of both, stress response and metabolic properties of intracellular STM persisters. Our analyses reveal that intracellular NR STM persister cells sense their environment and respond to stressors. NR and R STM can mount responses to stressors, and the low response observed for NR STM indicates that the persistent status of intracellular STM serves as protective niche.

## Results

**Design of fluorescent protein-based reporters for stress response of non-replicating intracellular *Salmonella enterica* at single cell level.** In this study, we aimed to analyze the stress response of NR, persisting intracellular STM at the single cell level. For this purpose, we changed the basic design of dual-fluorescence stress reporters[32] as shown in Fig. 1. Constitutive synthetic promoter $P_{EM7}$, derived from the bacteriophage T7 promoter, was replaced by the tet-ON cassette[33] to induce expression of a gene of interest (GOI) by addition of the non-antibiotic inducer anhydrotetracycline (AHT). The resulting dual-fluorescence reporters consisted of the tet-ON cassette for controlled expression of DsRed version DsRed T3_S4T[34], and sfGFP under control of the regulated promoters of SRS genes *msrA, trxA,* or *htrA* (Fig. 1A). Promoters of *msrA, trxA,* and *htrA* were selected to cover stress responses within the periplasm and the cytoplasm and because these showed stress-specific induction during our previous study[32].

The principle for the detection of persisters was adapted from Helaine et al.[13]. Addition of AHT to subcultures used as infection inoculum resulted in synthesis of DsRed. Before infection, AHT was removed by centrifugation and washing to terminate further DsRed synthesis. The principle of fluorescence protein (FP) dilution is that intracellular R STM continuously dilute DsRed, thus, decrease fluorescence levels, while NR or persistent STM maintain DsRed and consistent fluorescence levels. Furthermore, the sfGFP signal reported exposure to stressors and induction of SRS (Supplementary Fig. S1B).

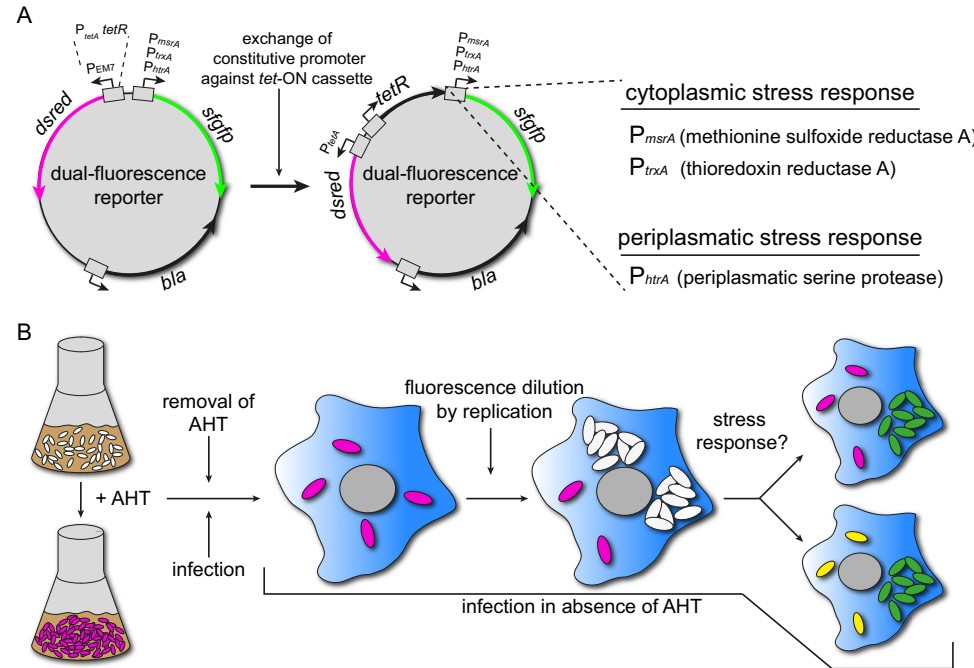

**Fig. 1 Design of reporters to measure stress response of non-replicating STM. A** For detection of NR intracellular STM, the EM7 promoter of dual FP reporters for *msrA*, *trxA*, or *htrA* was replaced by the tet-ON cassette to enable controlled induction of *dsred* expression by AHT. **B** Addition of AHT to growing cultures of STM induced DsRed synthesis. Before infection of macrophages, AHT was removed by centrifugation and washing. Infection was performed in absence of AHT. After infection, intracellular R STM dilute cellular DsRed levels. NR intracellular STM maintain cellular DsRed levels and were detected as DsRed-positive events. In addition, the *msrA*-induced sfGFP intensity provides information about response to stress imposed by the host cell.

To confirm function of newly generated reporter plasmids, we used in vitro cultures of STM WT harboring reporter [P*tetA*::*dsred* P*msrA*::*sfgfp*]. STM was grown in LB broth and samples were collected in hourly intervals for quantification of DsRed fluorescence levels by flow cytometry (FC) (Supplementary Fig. S1AB). Without further induction, DsRed intensity decreased over time proportional to bacterial replication, validating the principle of fluorescence dilution upon bacterial replication.

For infection of the murine macrophage-like cell line RAW264.7, STM WT harboring the reporter [P*tetA*::*dsred* P*msrA*::*sfgfp*] was grown overnight (o/n) in presence of AHT to induce expression of DsRed. RAW264.7 cells were infected by DsRed-positive STM WT. At 8 h p.i. in culture without AHT, the population was released, subjected to FC and the x-median RFI of DsRed was determined (Supplementary Fig. S1C). Two different subpopulations were detected when using STM WT [P*tetA*::*dsred* P*msrA*::*sfgfp*] (blue histogram). One subpopulation showed normal, non-diluted DsRed intensities comparable to constitutive DsRed expression via the EM7 promoter (Supplementary Fig. S1C, black histogram). The second subpopulation showed a lower DsRed intensity due to FP dilution. Without addition of AHT to o/n cultures, no DsRed-positive bacteria were detected (Supplementary Fig. S1C, magenta histogram). Furthermore, NR subpopulations were detected 16 h and 24 h p.i. using STM WT (Supplementary Fig. S1D), or various STM mutant strains. We also determined the detection accuracy of the cytometer used (Supplementary Fig. S1E–H). The measurements demonstrated that up to 10,000-fold differences in numbers of STM expressing DsRed or sfGFP were determined very precisely.

FC analysis of the distribution of DsRed and sfGFP fluorescence of the intracellular population revealed a very small DsRed-positive NR subpopulation (app. 0.4%), compared to the DsRed-negative R subpopulation (Supplementary Fig. S2A). We further determined the proportion of NR STM of various STM mutant strains. We introduced the reporters in mutant strains

unable to translocate SPI2-T3SS effector proteins (ΔssaV), or deficient in SPI2-T3SS effector proteins SifA or SseF thus unable (ΔsifA) or reduced (ΔsseF) in remodeling the host cell endosomal system[35]. Furthermore, ΔdksA was used as DksA is a transcriptional regulator activating repair of damages mediated by reactive nitrogen species. Mutant strains in *ssaV*, *sifA*, or *dksA* are highly attenuated in virulence and intracellular replication, whereas the *sseF* mutant strain shows moderate attenuation[36–39]. Calculating the proportion of NR STM demonstrated that ΔssaV and ΔdksA showed a higher proportion of NR STM of about 7% and 20%, respectively (Supplementary Fig. S1B), as a result of attenuated intracellular replication of R STM. Taken together, analyses of FP dilution allowed discrimination of R and NR STM, and detection of NR STM of mutant strains at various time points p.i.

**Stress response of non-replicating intracellular *Salmonella enterica*.** Next, we analyzed the stress response in NR compared to R STM. Since it is still matter of debate if NR persister cells are able to sense their environment, we aimed to investigate their ability to sense and react to stressors. For the detection of the entire intracellular population, STM WT harboring [P*EM7*::*dsred* P*xxx*::*sfgfp*] for constitutive DsRed expression was used. For detection of intracellular NR STM, STM WT harboring [P*tetA*::*dsred* P*xxx*::*sfgfp*] for AHT-induced DsRed expression was used. STM WT harboring the various reporters were used to infect RAW264.7 macrophages (Fig. 2). For STM WT harboring the reporters for detection of NR bacteria, AHT was added to o/n cultures of inoculum. The bacteria were released from host cells 8 h p.i. and subjected to FC (Fig. 2A, B), or immuno-stained against O-antigen and imaged by fluorescence microscopy (Fig. 2C–H). STM WT harboring a plasmid for constitutive expression of DsRed, but lacking expression of sfGFP served as negative control for adjustment of FC gating as described before[32].

For stress reporters P*msrA*::*sfgfp*, P*trxA*::*sfgfp*, P*htrA*::*sfgfp* investigated, we observed increased sfGFP levels in NR STM (Fig. 2A, B).

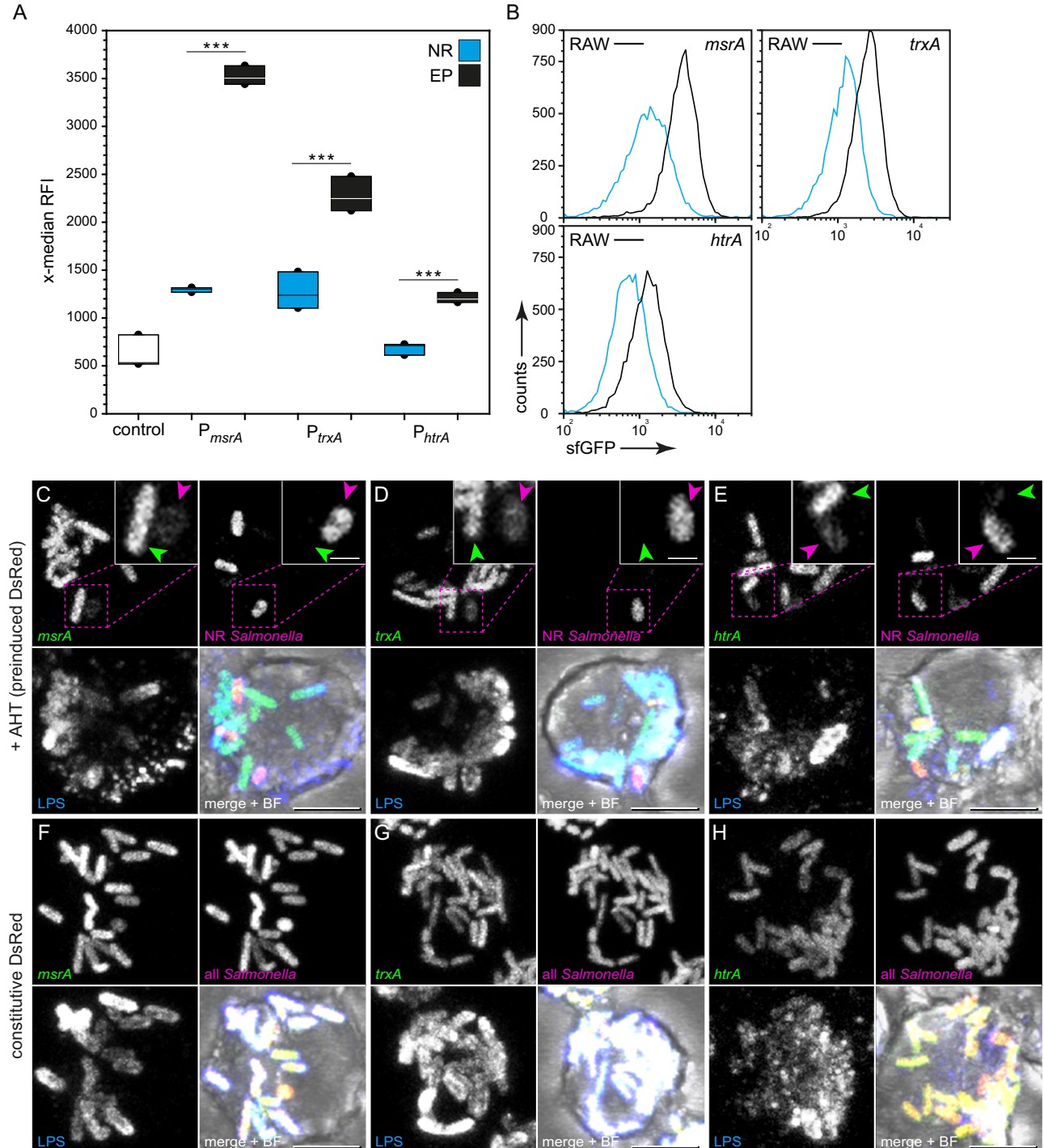

**Fig. 2 Lower levels of stress response of non-replicating STM WT compared to replicating intracellular STM.** STM WT harboring AHT-inducible (NR, blue bars) or constitutive (EP, black bars) dual-fluorescence reporter for *msrA, trxA* and *htrA* were grown o/n in LB medium containing 50 ng × ml$^{-1}$ AHT. Before infection of RAW264.7 macrophages, AHT was removed. At 8 h p.i., infected cells were lysed, or fixed for FC analyses (**A**, **B**), or microscopy (**C–H**), respectively. For FC, released STM were recovered, fixed, and analyzed. As non-induced control, dual FP reporter plasmids were used with a frameshift mutation in *sfgfp* (described before in the ref. [40]). **A** The x-median represents the stress-induced sfGFP signal of the DsRed-positive intracellular bacterial population 8 h p.i. Means and standard deviations of one representative experiment are shown. **B** Representative histograms corresponding to A are shown. Statistical analysis was accomplished by SigmaPlot by One Way ANOVA and significance levels are indicated as follows: *$p < 0.05$; **$p < 0.01$; ***$p < 0.001$; n.s. not significant. **C–H** The entire population of intracellular STM was detected by immuno-staining of O antigen (blue). **F–H** Constitutively *dsred*-expressing STM represent the entire intracellular bacterial population. **C–E** NR and R STM were positive or negative for DsRed fluorescence (magenta), respectively. sfGFP signals (green) indicate induction of *msrA, trxA*, or *htrA*. Representative NR STM and R STM are indicated by magenta and green arrowheads, respectively. Scale bars, 5 (overview), 1 μm (detail).

In contrast, the overall sfGFP expression of the entire population (EP) was higher compared to NR STM, i.e., 2.72-fold, 1.79-fold, and 1.77-fold increase of P$_{msrA}$::*sfgfp*, P$_{trxA}$::*sfgfp*, and P$_{htrA}$::*sfgfp*, respectively. Fluorescence microscopy showed comparable results (Fig. 2C–H). R and NR STM were readily distinguished by the absence or presence of DsRed fluorescence, respectively (Fig. 2C–E). Correlation of DsRed to sfGFP fluorescence signals showed lower induction of *msrA, trxA,* or *htrA* in NR STM. STM WT harboring stress reporters with constitutive DsRed expression all displayed DsRed and sfGFP fluorescence, indicating induction of *msrA, trxA,* or *htrA* (Fig. 2F–H). The inoculum without AHT addition lacked DsRed synthesis, and accordingly DsRed fluorescence was not detectable (Supplementary Fig. S3). NR STM WT resided in LAMP1-positive compartments throughout intracellular presence (Supplementary Fig. S4), in line with prior findings[13] and proving precision of our dual-fluorescence reporter system. We further checked stress induction of R and NR STM after inhibition of NADPH oxidase of RAW macrophages. For this, we treated host cells with 10 µM diphenyleneiodonium chloride as previously described[40], and investigated *msrA* and *trxA* induction of R and NR STM 8 h p.i. (Supplementary Fig. S3G). After inhibition of reactive oxygen species production, we detected lower sfGFP levels for the entire population. In contrast, the sfGFP level of the NR population did not change after inhibition of reactive oxygen species production and remained at low level.

We have reported previously that specific virulence factors of STM contribute to reduce exposure to host defense mechanisms and stressors[32]. Therefore, we compared the stress response of R STM to NR STM strains deficient in SPI2-T3SS or SRS in RAW264.7 macrophages at various time points p.i. (Fig. 3). We used P$_{msrA}$ as representative for induction of SRS. For all mutant strains analyzed, we observed a lower *msrA* induction of the NR subpopulation compared to the entire population at 24 h p.i. (Fig. 3A). The same trend was observed using the reporters for *htrA* and *trxA* (Supplementary Fig. S3H, I). We further calculated the time-resolved stress response of R and NR STM in order to investigate continuous stress-exposure over time of infection (Fig. 3B). The depicted slopes represent the sfGFP intensity increase over the time of infection (8–24 h p.i.). Higher stress induction over a constant period of time resulted in a higher slope as shown for example for the entire population of STM ΔssaV and ΔdksA strains (magenta and yellow line in Fig. 3B). We observed that stress induction in NR STM only increases very slightly during the course of infection (dashed lines). In contrast, R STM showed a high increase of the sfGFP signal.

We can conclude that intracellular NR STM sense external stress conditions and did not encounter the same stress levels as R STM.

**Metabolic activity analyses of intracellular persisters at the early phase of infection.** The lower induction of stress reporters in NR STM can be explained by (i) lower levels of stressor acting on this population compared to R STM, or (ii) reduced biosynthetic activity resulting in lower synthesis of sfGFP. To distinguish and to check that the lower induction of reporters is not erroneously a consequence of a reduced biosynthetic activity, we analyzed the ability of reporter protein biosynthesis (further referred as metabolic activity) of NR STM WT. For this, dual-fluorescence reporter plasmid harboring arabinose-inducible DsRed expression and AHT-inducible sfGFP expression was generated (Supplementary Fig. S5A). Presence of arabinose during culture of inoculum resulting in synthesis of DsRed. After removal of arabinose and infection of RAW264.7 macrophages, R STM dilute DsRed while NR STM maintain DsRed levels. sfGFP expression was induced by addition of AHT to infected cells

serving as proxy for the biosynthetic capacity, and thus metabolic activity of STM (Supplementary Fig. S5B, C). For functional control, STM WT harboring the double-inducible dual-fluorescence reporter [P$_{BAD}$::*dsred* P$_{tetA}$::*sfgfp*] was grown in LB medium and samples were collected for quantification of DsRed and sfGFP levels by FC (Supplementary Fig. S5D–F). Arabinose was added at the beginning of subculture (−4 h). After 4 h, the x-median RFI of DsRed increased, however, the sfGFP intensity remained constantly low. After removal of arabinose and start of a further subculture without the presence of arabinose but in the presence of AHT (0 h), the intensity of DsRed decreased within 3 h of subculture due to fluorescence dilution. In contrast, the x-median RFI of sfGFP increased (+3 h), confirming induction by AHT.

Next, we analyzed the metabolic activity of NR STM WT [P$_{BAD}$::*dsred* P$_{tetA}$::*sfgfp*] in RAW264.7 macrophages at 24 h p.i. (Fig. 4) following the experimental design depicted in Supplementary Fig. S5C. At 22 h p.i., AHT was added directly to the cell culture medium to induce expression of sfGFP. At 24 h p.i., bacteria were released from host cells and subjected to FC as described above. Plotting the population against their DsRed and sfGFP intensities allowed clear discrimination of the various subpopulations (Fig. 4A). STM positive only for DsRed represent metabolically inactive NR persisters. STM positive both for DsRed and sfGFP represent metabolically active NR persisters. STM only sfGFP-positive represent a metabolically active R population. Particles both negative for DsRed and sfGFP represent the background signal containing host cell debris. Without the addition of arabinose to o/n cultures, and without addition of AHT to infected cells, neither DsRed-positive nor sfGFP-positive STM was detected by FC (Fig. 4A). Comparing the amount of metabolically active to inactive NR STM revealed that app. 30–40% of all persisters showed metabolic activity at 24 h p.i. (Fig. 4B). Comparing the metabolic activity of active persisters to active R STM showed that the metabolic capacity did not differ significantly (Fig. 4C), demonstrating that metabolically active NR STM showed the same metabolic activity compared to R STM.

However, if about half of the persistent population was inactive at 24 h p.i., why did we only detect a homogeneous stress-induced population instead of two subpopulations? (Fig. 2B). Is there one subpopulation that showed stress responses and another lacking response? To measure the metabolic activity at 24 h p.i., AHT was added 22 h p.i. If a persister is metabolically active at 22 h p.i., this cell starts sfGFP synthesis (Fig. 4D), but inactive persisters do not. However, a persister metabolically inactive at 22 h p.i. may have been active in the prior period of infection (Fig. 4E). If this holds true, the proportion of active persisters should be high at beginning of infection and decrease over time of infection, similar to result by Helaine et al.[15] for time points 24–72 h p.i. Since stress induction is measured by accumulation of sfGFP, an active persister harboring the dual-fluorescence stress reporter may synthesize sfGFP at the early phase of infection, become inactive e.g., at 10 h p.i., thus appears metabolically inactive at 24 h p.i. However, such cell would still contain sfGFP synthesized in response to stressors, and should be detected as stress signal-positive at 24 h p.i. (Fig. 4E).

To test this hypothesis, we calculated the proportion of metabolically active and inactive persisters in the early phase of infection (Fig. 4G, H, FC histogram shown in Fig. 4F). For that, AHT was added at indicated time points, however, host cells were lysed 24 h p.i. for clear separation of R and NR STM. At 2 h p.i., appr. 95% of all NR STM were metabolically active. Starting 8 h p.i., the amount of active persisters decreased constantly. The metabolic capacity of this active persistent population was always on the same level compared to active R STM (Fig. 4I). However, in the early phase of infection from 2 to 10 h p.i., the metabolic capacity of

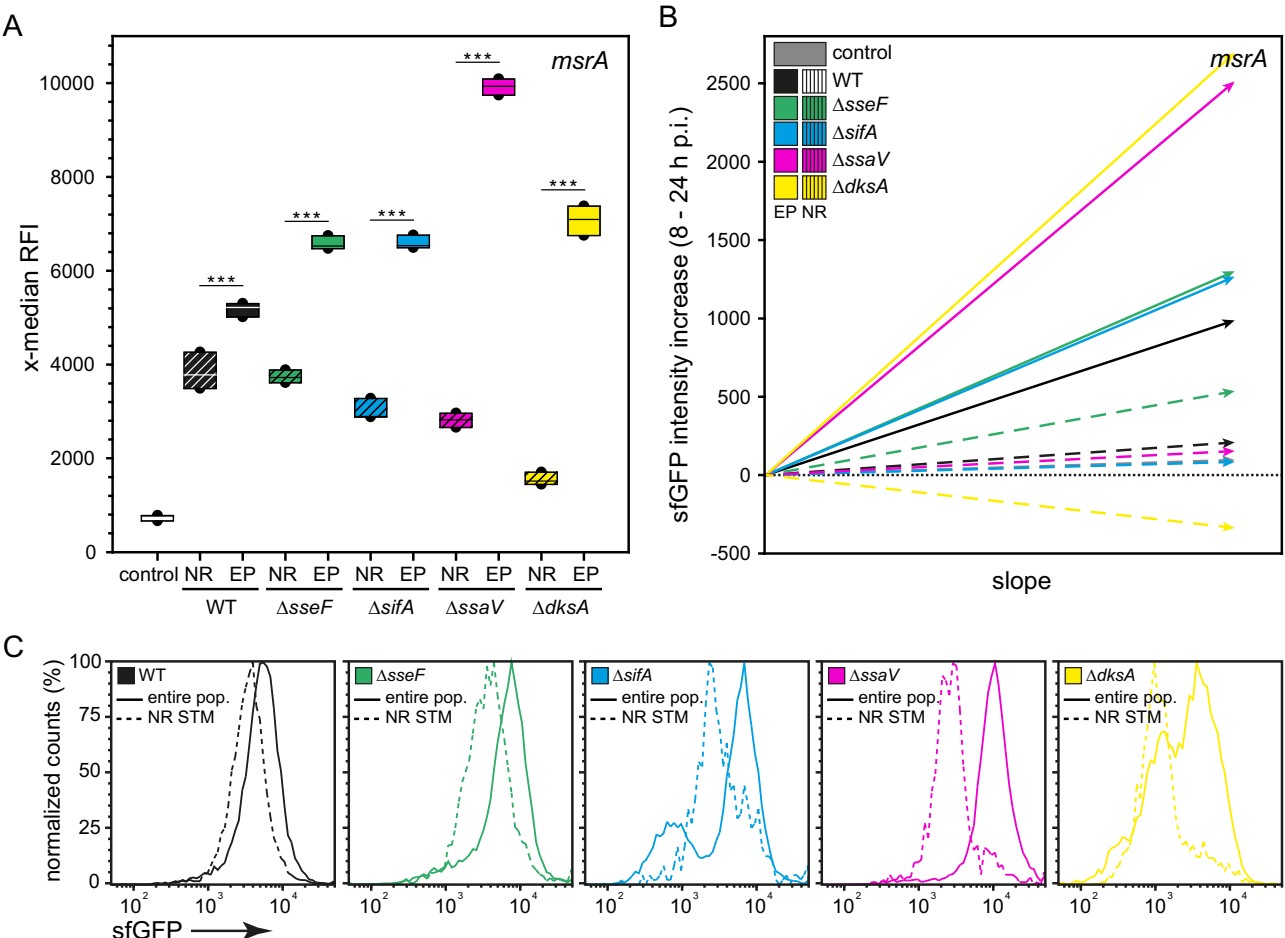

**Fig. 3 Non-replicating SPI2-T3SS or SR mutant strains also show lower stress response compared to respective subpopulation.** STM WT, $\Delta sseF$, $\Delta sifA$, $\Delta ssaV$, and $\Delta dksA$ strains harboring AHT-inducible (NR), or constitutive (entire population, EP) dual-fluorescence reporter for $msrA$ were grown o/n in LB medium. AHT was present for AHT-inducible reporters and removed before infection. RAW264.7 macrophages were infected, lysed 24 h p.i., liberated STM were recovered, fixed, and subjected to FC analyses. As negative control, a dual FP reporter plasmid with $sfgfp$ inactivated by a frameshift was used. **A** The x-median represents the $msrA$-induced sfGFP signal of the DsRed-positive intracellular bacterial population at 24 h p.i. Means and standard deviations of one representative experiment are shown. Statistical analysis was performed as described for Fig. 2 and is indicated for NR vs. EP for each strain. **B** The depicted slopes represent the sfGFP intensity increase over the time of infection. **C** Representative histograms corresponding to **A** are shown.

active NR STM was even higher compared to active R STM. The higher x-median RFI of sfGFP of both, active R and NR STM at the early time points p.i. was resulting because all samples were lysed 24 h p.i.

We also calculated the metabolic capacity of the entire NR STM population, because when measuring stress induction of NR STM we are not able to discriminate between metabolically active and inactive NR STM (Fig. 4J, K). In addition, we calculated the ratio of the metabolic activity of all NR compared to active R STM (Fig. 4L). Values above a ratio of 1 (red line) indicate higher metabolic activity of the entire NR population. The metabolic activity of the entire NR population up to 10 h p.i. was approximately at the same level as that of R STM. To conclude, because of the same activity of R and NR STM, the lower stress response of persisters 8 h p.i. is not due to a lower metabolic capacity of persisters.

**Antibiotic exposure of persisters results in decreased stress response.** Since intracellular persisters are able to withstand bactericidal antibiotic exposure, we exposed STM in RAW264.7

macrophages to cefotaxime, a cephalosporin affecting bacterial cell wall synthesis. We measured stress induction of NR STM WT at 24 h p.i. As representative indicator for stress response induction $P_{msrA}$ was used. At 10 h p.i., 200 μg × ml$^{-1}$ cefotaxime was added, following incubation for further 14 h. After cefotaxime treatment for 14 h, the subpopulation of R STM was highly reduced (Supplementary Fig. S6A). The proportion of NR STM WT increased from 0.32 to 13% (Supplementary Fig. S6B) representing a 41.04-fold increase (Supplementary Fig. S6C). However, the replicating population has not been fully removed, in contrast to prior observations[15]. Fluorescence microscopy of infected RAW264.7 macrophages at 24 h p.i. (Supplementary Fig. S6D, E) showed signal patterns supporting the FC data. Without antibiotic treatment, STM WT replicated efficiently in macrophages also containing NR STM showing DsRed fluorescence (Supplementary Fig. S6E).

After cefotaxime treatment, RAW264.7 macrophages contained both, R and NR STM WT (Supplementary Fig. S6E, middle panel, indicated by magenta and green arrows), or NR STM WT only (Supplementary Fig. S6E, lower panel). Analysis of stress response showed that after antibiotic treatment the

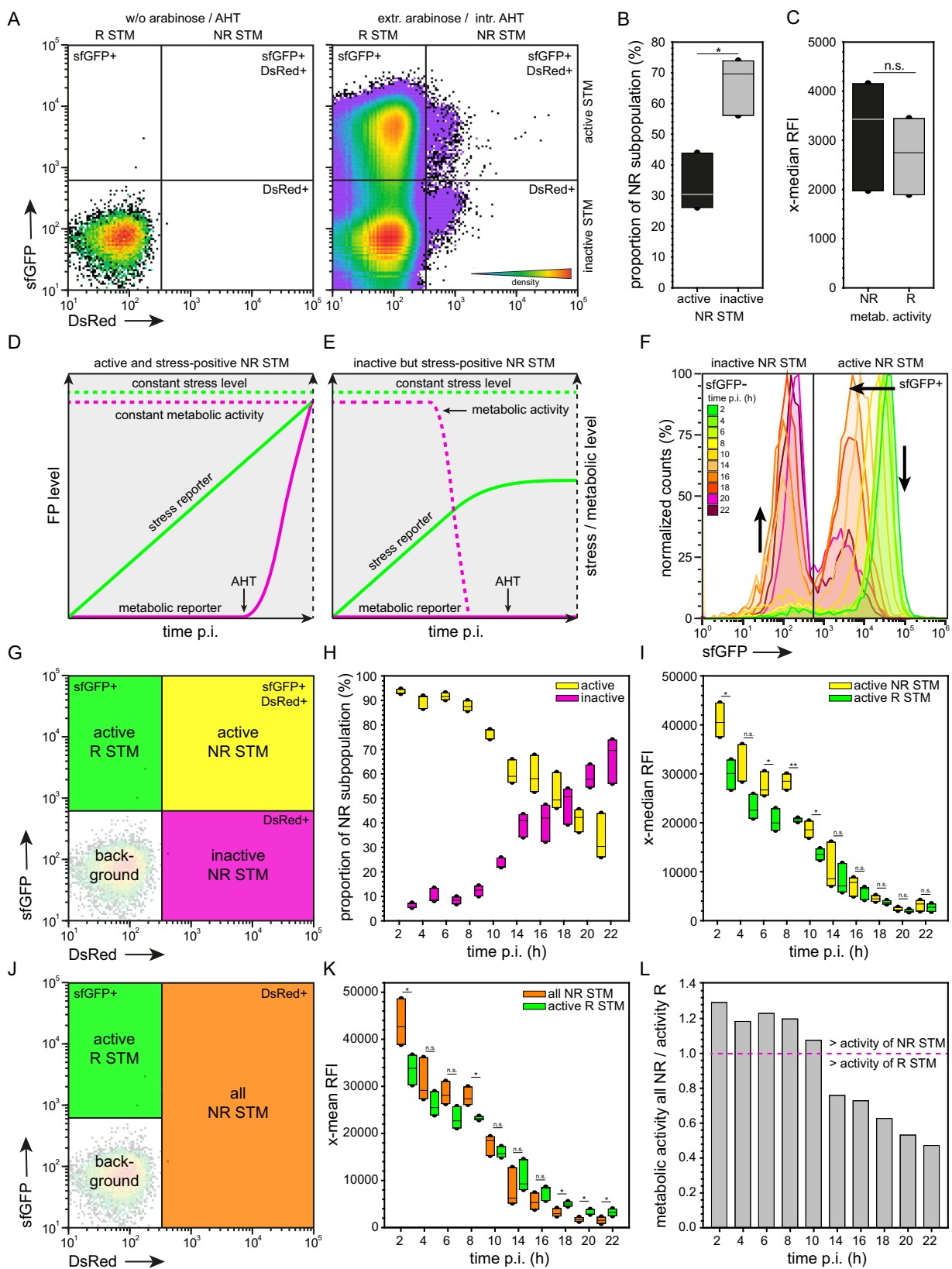

x-median RFI of sfGFP decreases, however, cefotaxime-treated NR STM WT still showed an active stress response (Fig. 5A). When we checked the entire intracellular population, we observed that after cefotaxime treatment the *msrA* induction also dropped to a level comparable to non-treated NR STM.

Since persisters are a subpopulation of NR STM that are able to resume growth after release from host cells, we controlled that we indeed analyzed growth-competent persister cells. For that, we inoculated the released population after lysis of macrophages into fresh LB medium and calculated the relative amount of DsRed-

**Fig. 4 Non-replicating persisters show equal metabolic activity compared to replicating STM WT in the early phase of infection.** STM WT harboring double-inducible dual-fluorescence reporter was grown o/n in LB medium in the presence of arabinose. Before infection, arabinose was removed. RAW264.7 macrophages were infected, AHT was added 22 h p.i., and cells were lysed 24 h p.i. Liberated STM were recovered, fixed and subjected to FC. **A** Plotting the entire intracellular bacterial population against their arabinose-induced DsRed and AHT-induced sfGFP intensity shows distinct populations. As control, STM without addition of arabinose or AHT was analyzed. **B** The proportion of metabolically active and inactive NR STM WT at 22 h p.i. is shown. **C** Comparison of the metabolic activity of active NR and R STM WT 22 h p.i. **D–H** STM WT was cultured, infected and prepared for FC as described above. AHT was added at various time points p.i. and cells were lysed 24 h p.i. **D** FP levels anticipated for metabolically active and stress signal-positive NR STM. **E** Detected metabolically inactive NR STM may be stress signal-positive because of metabolic activity before addition of AHT. **F** Representative histograms of the metabolic activity of intracellular NR STM WT at various time points p.i. Arrows indicate increase in metabolically inactive NR STM, and decrease in metabolically active NR STM over time of infection. **G** Yellow, green, and magenta gates define subpopulations compared in **H** and **I**. **H** The proportion of metabolically active and inactive NR STM at the various time points was calculated. **I** Comparison of the metabolic activity of active NR and R STM WT at various time points p.i. **J** Green and orange gates define subpopulations compared in **K**. **K** Comparison of the metabolic activity of all NR and R STM WT at various time points p.i. **L** The ratio of metabolic activity of all NR STM to metabolic activity of R STM is shown. Means and standard deviations of one representative experiment are shown. Statistical analysis was performed as described for Fig. 2.

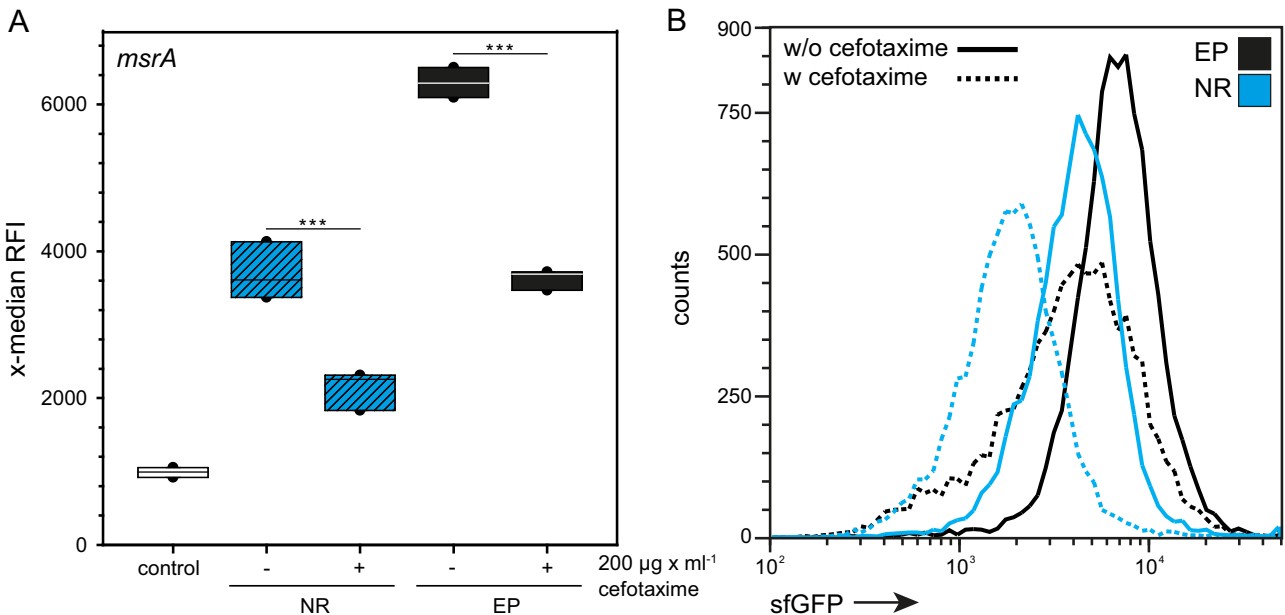

**Fig. 5 Cefotaxime-treated non-replicating STM WT show slight induction of stress response.** STM WT harboring AHT-inducible (NR) or constitutive (EP) dual-fluorescence reporter for *msrA* was grown o/n in LB medium, in the presence of AHT or the NR reporter. Before infection, AHT was removed. RAW264.7 macrophages were infected, cefotaxime was added to the cells at 10 h p.i. if indicated, and cells were lysed at 24 h p.i. Liberated STM were recovered, fixed and subjected to FC. As negative control, a dual FP reporter plasmid with *sfgfp* inactivated by a frameshift was used. The x-median (**A**) represents the $P_{msrA}$-induced sfGFP signal of the DsRed-positive intracellular bacterial population, and **B** shows a representative histogram. Means and standard deviations of one representative experiment are shown. Statistical analysis was performed as described for Fig. 2.

positive persisters 0, 2, 4, and 6 h after reinoculation (Supplementary Fig. S7A). The number of DsRed-positive events (NR persisters) did not decrease within the first 4 h after reinoculation (Supplementary Fig. S7B, C). After that, the number of DsRed-positive events decreased to about 50%, indicating that NR persisters are still viable and competent to regrow. As determined in Supplementary Fig. S1E–H, ratios of up to 1:10,000 between STM DsRed and STM sfGFP were precisely determined. In addition, detection of the optical density of the reinoculated cultures showed that cefotaxime-treated STM WT started to replicate after 4–6 h (Supplementary Fig. S7D). Therefore, the persisters analyzed were able to regrow 6 h after reinoculation that was in line with optical density of cultures.

**Stress response of persisters within primary human phagocytes.** Finally, we determined the stress response of NR STM within primary human monocytes to compare stress-induction of

NR STM between murine and human phagocytes. Primary cells were isolated from buffy coat and showed M1 polarization after differentiation. $P_{msrA}$ was used as representative for stress response induction. Infection and FC analysis was performed as described above and NR STM were detected inside human macrophages at 8 h p.i. (Fig. 6A). Comparison of *msrA* induction of NR STM in murine and human macrophages revealed an equal x-median RFI of sfGFP (Fig. 6B). In addition, the *msrA* induction of the NR subpopulation [$P_{tetA}$::*dsred* $P_{msrA}$::*sfgfp*] and the entire population [$P_{EM7}$::*dsred* $P_{msrA}$::*sfgfp*] showed the same level in human macrophages. Plotting the bacterial population against their AHT-induced DsRed and *msrA*-induced sfGFP intensity showed that in primary human macrophages only NR STM were present, while a subpopulation of R STM was absent (Fig. 6C). To conclude, also in primary human macrophages NR STM WT showed an active stress response that was comparable to RAW264.7 macrophages.

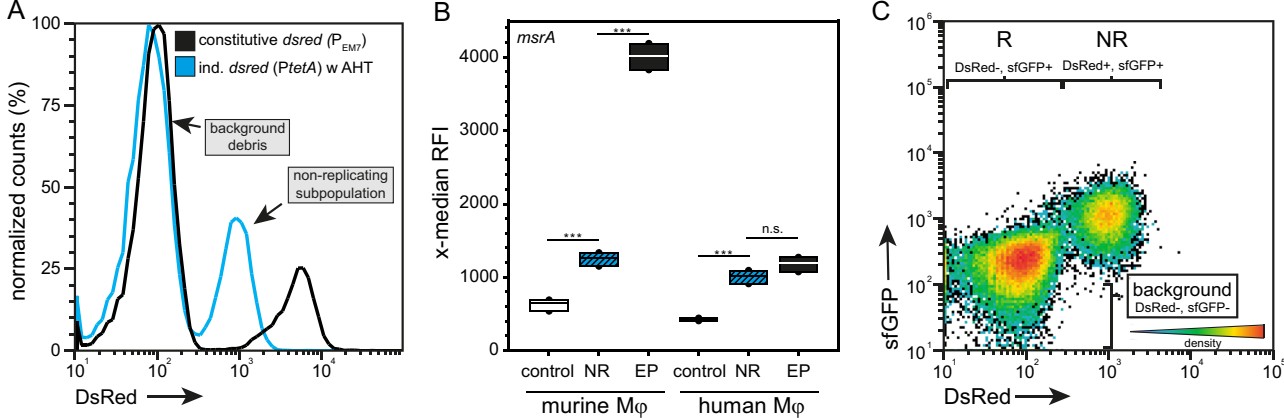

**Fig. 6 Stress response of non-replicating STM WT in primary human macrophages.** STM WT harboring AHT-inducible (NR) or constitutive (EP) dual-fluorescence reporter for *msrA* was grown o/n in LB medium in the presence of AHT if necessary. Before infection, AHT was removed and murine RAW264.7 or human primary macrophages were infected. Host cells were lysed 24 h p.i., liberated STM were recovered, fixed, and subjected to FC. As negative control, a dual FP reporter plasmid with *sfgfp* inactivated by a frameshift was used. **A** Detection of intracellular NR STM in human macrophages. **B** Comparison of *msrA* induction within murine and human macrophages. The x-median represents the *msrA*-induced sfGFP signal of DsRed-positive intracellular bacteria. Means and standard deviations of one representative experiment are shown. **C** STM WT only shows a NR subpopulation when plotting the bacterial population against their AHT-induced DsRed and *msrA*-induced sfGFP intensity. Statistical analysis was performed as described for Fig. 2.

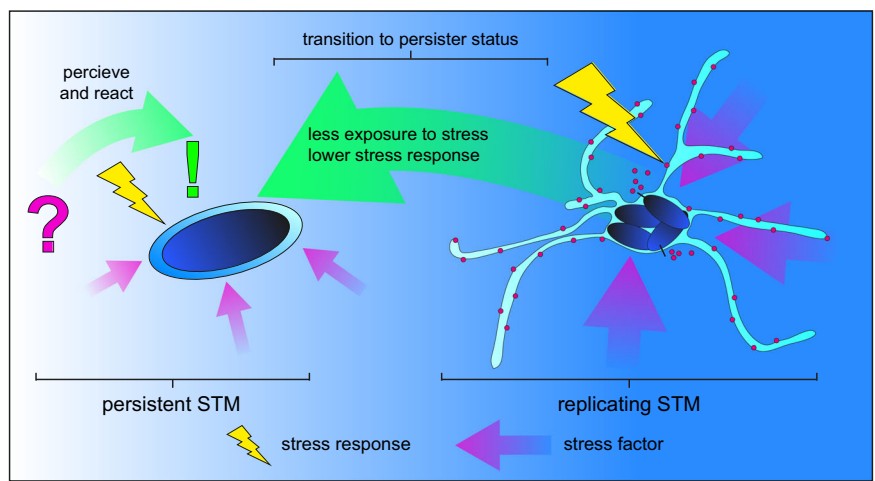

**Fig. 7 Transition to persister status results in decreases exposure to stressors and enables survival of harsh conditions in host cells.** Replicating intracellular STM show diverse stress responses in reaction to harsh environmental conditions. Intracellular persistent STM are able to sense and respond to stressors by turning on SRS. The induced stress response of persisters is lower compared to the replicating subpopulation of intracellular STM suggesting that persisters encounter lower exposure to stressors. The transition to a persistent state inside host cells promotes stress tolerance, survival, and dissemination of the pathogen due to re-growth after subsidence of stressful conditions.

## Discussion

**NR intracellular STM are capable to respond to host-imposed stressors.** In this study, we further developed and applied FP-based reporters for analysis of stress response of intracellular NR STM at single cell level. This approach enables quantification of the stress response of distinct intracellular subpopulations of STM. Our study demonstrates that NR STM persister cells sense their intracellular environment, and respond to stressors by activating a stress response even under antibiotic pressure (Fig. 7). These cells regrew in rich medium after infection, proving that proper NR STM persister cells were examined.

Prior work reported that SPI2-T3SS-mediated intracellular survival was abrogated in γ-Interferon-activated, reactive nitrogen species-producing macrophages, but also indicated STM survival in absence of SPI2-T3SS function[40]. Our findings support previously reported metabolic activity of intracellular STM persisters, and translocation by SPI2-T3SS[13,15,16]. STM persisters

express SPI2 genes and for this, bacteria must sense their environment to activate gene expression[16]. The overall number of persisters with active SPI2-T3SS may be very low, but these bacteria could critically contribute to the maintenance of the STM population. Our results corroborate that STM persister cells are not segregated from the intracellular habitat, but rather sense their environment and changes within. Persisters that maintain effector delivery and are able to respond to stress factors but cease to grow, still provide evolutionary benefit. Since the persister status is reversible and a single cell can give rise to a new susceptible population, the intracellular heterogeneity of *Salmonella* provides adaptive advantage and favors survival of the entire population during infection of host cells or in general, under adverse environmental conditions. Since we observed lower stress response of persister cells compared to R STM, independent from the duration of infection or function of virulence factors, we conclude that the persistent status serves as a kind of niche by

protecting NR STM from severe stress exposure (Fig. 7). The host cell may no longer be able to exert stress on NR STM persister cells, which in turn results in lower stress response. Beyond this, STM persisters reside individually within segregated *Salmonella*-containing vacuoles during the whole course of infection, further promoting their protective niche. This compartment shields STM from the cytosolic environment and host cell defense mechanisms such as xenophagy reviewed in the ref.[41]. In addition, the *Salmonella*-containing vacuole of NR STM is separated from the *Salmonella*-containing vacuole/*Salmonella*-induced filament continuum containing R STM[35].

**At initial infection, a high proportion of NR STM is metabolically active**. We observed that at initial infection of macrophages, the proportion of metabolically active NR STM was as high as 95%. Thus, the lower stress response of NR STM is not a consequence of a lower metabolic capacity. Starting about 8 h p.i., the proportion of active persisters constantly decreased. This is in line with previous observations that detected about 70% of active NR STM in RAW264.7 macrophages 12 h p.i., and about 25% active NR STM at 24 h p.i[13]. Our studies extended these analyses to early time points p.i. In addition, we compared the level of metabolic activity, which was comparable to R STM until 10 h p.i., if we average the total activity of the entire NR population. If only metabolically active NR STM persisters are considered, these did not show a significantly different activity compared to active R STM over the entire period of infection. If a small reservoir of persisters always remains active over a longer period of time, this may allow formation of new populations in the host, and thus to trigger relapses. The metabolic activity of NR STM at the beginning of infection may serve to successfully establish their persistent status. For *M. tuberculosis* or *E. coli*, in vitro transcriptome studies indicated that persisters showed downregulation of metabolic genes and therefore decreased metabolism[42,43]. Genes with functions in energy production and tricarbonic acid cycle were downregulated in *Burkholderia cenocepacia* persisters[44]. For intracellular STM, however, there is evidence that distinct subpopulations of metabolically active and inactive persisters exist[13,16]. The observation of metabolically active NR bacteria is not limited to *Salmonella* species, but also reported for *M. tuberculosis* in macrophages, and could therefore be a widespread phenomenon[14].

When examining persistent bacteria, it is highly important which time point is analyzed, and to be aware that overall metabolic activity of the NR population changes over time. This refers to the decreasing proportion of metabolically active persisters over time, and the associated activity of the entire population on average. In many studies, late time points were selected to investigate persisters, at which a small proportion of metabolically active persisters was detected, which further decreased[13]. Our studies are in line with these results and our data also showed that when the entire NR subpopulation is considered, a decreased metabolic activity from 14 h p.i. was observed. This is also in line with a model of reduced metabolism in persister cells. Importantly, before 14 h p.i. the entire population of NR STM persisters on average does not show a reduced metabolism and the metabolic activity is not lower at any time point analyzed, if only metabolically active NR STM are considered. However, since the proportion of metabolically active persisters decreases over time, the metabolic activity of the entire population of NR STM on average decreases accordingly. In turn, a reduced metabolic activity is measured when the entire population of NR STM is observed. Conventional transcriptome analyses or other population-wide approaches average the entire NR population and fail to distinguish between active and inactive persisters. Single cell analyses that are capable of addressing small subpopulations of intracellular STM, such as the approaches applied here, provide a more precise insight into the physiological state of persisters. Novel single cell transcriptomics approaches for bacteria are emerging[45], and may provide a future option for analyses of persisters.

**Stress response of NR STM in non-permissive primary human macrophages**. We investigated stress response of NR STM persisters in much less permissive human phagocytic cells. We observed that the level of stress response triggered in human macrophages was comparable to that of STM in RAW264.7 macrophages being rather permissive for STM intracellular proliferation. These results suggest that the induced stress response of NR STM is either independent of the killing capacity of the host cell, or because persisters are so well protected that even phagocytes with high antimicrobial activity do not induce higher levels of stress response.

Typhoidal serovars (TS) of *S. enterica* cause systemic infections in humans such as typhoid fever, and persistent or recurrent forms of the disease are frequent[46]. Advanced analyses of stress response shall be extended to persister cells of typhoidal serovars Typhi and Paratyphi A to gain insights into their physiological state, and to test the ability to sense stress factors. As part of the "stealth strategy"[47], persister formation in typhoidal serovars may occur at higher frequency. In relation to heterogeneity, it will be of interest if intracellular Typhi and Paratyphi A show distinct subpopulations with different levels of stress response. Knowledge of persister cell frequency, the proportion of metabolically active and inactive persisters, and the level of stress response of typhoidal *Salmonella* in comparison to STM may lead to new insight into patho-mechanisms of typhoidal *Salmonella* and recurrent infections. This will require further analyses in other cell lines like human phagocytic cell line U937 or primary phagocytic cells.

**Stress response of persisters and new therapeutic options?** Since persisters that emerge during infection are extremely well protected and the clinical relevance of relapsing infections due to persisters is high, new strategies to eliminate persister cells are of utmost therapeutic importance. Current approaches include prevention of persister formation, identification of antimicrobial compounds that act on persister cells, or resuscitation of persister cells to restore sensitivity to conventional antibiotics reviewed in the ref. [48]. Promising studies have been performed to identify compounds acting on bacterial persister cells, however further action is required. High-throughput screening methods are often applied, but many screens were performed using planktonic persisters in vitro. Recent work showed that the efficacy of drugs against tuberculosis is significantly different when applied in vivo compared to in vitro conditions[49]. This may also apply to other drugs, making analyses of effects on persister cells during interplay with their host essential. One approach could be host-directed therapies focusing on host–pathogen interactions, as already has been demonstrated for *M. tuberculosis* infections by treating infections with small molecules to interfere with host responses or persistence[50,51]. Antibody–antibiotic conjugates are another emerging approach. Antibiotics are coupled to antibodies against a specific pathogen to increase the efficiency of antibiotics against intracellular pathogens including persisters[52]. Zhou et al.[53] showed that *S. aureus* bacteremia in mice was successfully treated, hence, this approach may serve as a novel therapeutic platform in the future. Our finding of the continuing function of SRS in persister cells and the ability of persisters to sense stressors may lead to new options for resuscitation of persisters examples

in the ref. [54]. If deregulation of SRS by decoy compounds is possible, this may lead to re-initiation of normal growth and increased antibiotic susceptibility. Approaches to interfere with quorum sensing for interbacterial communication reviewed in[55] or biofilm matrix production[56] enabled new antimicrobial strategies. In contrast, potential interference with SRS in persisters needs to target single cells in complex populations within host organisms.

**Conclusions and outlook**. In summary, we have developed and applied new dual FP reporters that enable extremely sensitive analyses of stress response and metabolic activity of NR persister cells. Of particular interest is that NR STM sense their environment and display a lower stress response with constant metabolic activity during the early phase of infection. Our findings provide deeper knowledge that besides the ability to subvert immune defenses of the host, NR STM persister cells maintain capability to sense stressors and react to stress. These persisters exhibit constant metabolic activity at the beginning of the infection, which supports the pathogen to create a more permissive environment for recrudescent infections.

## Methods

**Generation of reporter plasmids**. For the generation of a reporter plasmid to detect non-replicating STM, plasmid p5084 ($P_{EM7}$::dsred $P_{msrA}$::sfgfp), p5085 ($P_{EM7}$::dsred $P_{trxA}$::sfgfp), and p5055 ($P_{EM7}$::dsred $P_{htrA}$::sfgfp) with a constitutive expression of DsRed and msrA, trxA, or htrA regulated expression of sfGFP was used. Via Gibson Assembly of PCR fragments, the EM7 promoter was replaced by the tet-ON cassette to enable artificial induction of DsRed by anhydrotetracycline (AHT).

For the generation of a single FP reporter plasmid to detect non-replicating STM, plasmid p5204 ($P_{EM7}$::dsred $P_{cypD}$::sfgfp (frameshift)) with a constitutive expression of dsred and a frameshift mutation in sfgfp was used. Via Gibson Assembly, the EM7 promoter was replaced by the tet-ON cassette to enable artificial induction of dsred by AHT.

For the generation of a dual-fluorescence vitality reporter plasmid to detect the metabolic activity of non-replicating STM, plasmid p4928 ($P_{EM7}$::tagrfp-T $P_{tetA}$::sfgfp) with a constitutive expression of tagrfp-T and an AHT-inducible expression of sfgfp was used. Via GA, the EM7 promoter was replaced by araC $P_{BAD}$ cassette to enable artificial induction of tag-rfpT by arabinose. Subsequently, p5419 ($P_{BAD}$::tagrfp-T $P_{tetA}$::sfgfp) was used to amplify the $P_{BAD}$ and tet-ON cassette. Via GA, $P_{EM7}$ and the msrA promoter of p5084 ($P_{EM7}$::dsred $P_{msrA}$::sfgfp) was replaced by the $P_{BAD}$ and tet-ON cassette to enable artificial induction of dsred by arabinose and sfgfp by AHT. The resulting single and dual FP reporter plasmids are listed in Supplementary Table S1 and oligonucleotides used for construction are listed in Supplementary Table S3.

**Bacterial strains, growth conditions AHT, and arabinose induction**. Salmonella enterica sv. Typhimurium strain NCTC12023 (STM) was used as wild-type strain and isogenic mutant strains used in this study are listed in Supplementary Table S2. Bacteria were cultured in lysogeny broth (LB) at 37 °C overnight (o/n) using a roller drum at 60 rpm with aeration. For maintenance of plasmids, carbenicillin was added at 50 μg × ml[−1] as a selection marker. For induction of expression of $P_{tetA}$-controlled dual FP reporter (p5205, p5300, p5202, or p5418) or $P_{BAD}$-controlled dual FP reporter (p5426) AHT or arabinose always was directly added to LB to a concentration of 50 ng × ml[−1] or 13.3 mM, respectively, in the o/n culture and removed when necessary by centrifugation for 3 min at 5000×g and washing with fresh LB.

**Isolation of primary human monocytes and differentiation**. Buffy coat was obtained via the blood bank of Deutsches Rotes Kreuz (Springe, Germany) from pooled samples of voluntary, anonymous blood donors who gave informed consent. Lymphocytes were prepared from buffy coat by Ficoll-Hypaque density gradient centrifugation[57]. Buffy coat was diluted in PBS (ratio 1:3) and Ficoll-Hypaque was added, following centrifugation for 20 min at 800×g. Afterwards, $5 × 10^8$ cells were cultured in Roswell Park Memorial Institute (RPMI) medium containing 5.5 g × l[−1] NaCl, 5.0 mg × l[−1] phenol red, 2.0 g × l[−1] NaHCO₃, 25 mM HEPES, 4 mM stable glutamine without sodium pyruvate (Biochrom), 100 units × ml[−1] penicillin, 100 μg × ml[−1] streptomycin, 2.5 ng × ml[−1] GM-CSF (granulocyte-macrophage colony-stimulating factor) and supplemented with 10% human plasma. After o/n incubation at 37 °C in an atmosphere of 5% CO₂ and 90% humidity, non-adherent cells were removed, and adherent cells were cryo-preserved in inactivated fetal calf serum (iFCS) and DMSO. Next, cell suspensions were thawed, seeded, and maintained in RPMI-1640 (Biochrom),

supplemented with 20% FCS and 2.5 ng × ml[−1] GM-CSF (Peprotech) to differentiate monocytes into macrophages. After 5–7 days, the purity of the macrophage population was checked by staining with FITC anti-human CD14 antibody (BioLegend) and FC analyses. The macrophages were used for infection by STM strains.

**Cell lines and cell culture**. For infection experiments murine RAW264.7 macrophages (American Type Culture Collection, ATCC no. TIB-71), RAW264.7 macrophages stably transfected with LAMP1-GFP or primary human monocytes were used. RAW264.7 macrophages were cultured in Dulbecco's modified Eagle's medium (DMEM) containing 3.7 g × l[−1] NaHCO₃, 4.5 g × l[−1] glucose, 4 mM stable glutamine without sodium pyruvate (Biochrom) and supplemented with 6% iFCS (Sigma-Aldrich). Primary human monocytes were cultured in RPMI medium containing 5.5 g × l[−1] NaCl, 5.0 mg × l[−1] phenol red, 2.0 g × l[−1] NaHCO₃, 25 mM HEPES, 4 mM stable glutamine without sodium pyruvate (Biochrom) and supplemented with 10% iFCS (Sigma-Aldrich).

**Host cell infection (gentamicin protection assay) for cytometry**. Before infection, RAW264.7 or primary human macrophages were seeded in surface-treated 6-well plates (TPP) to reach confluency (∼2 × 10⁶ cells per well) on the day of infection. For infection, Salmonella strains were grown o/n in LB broth (app. 18 h). Infection was performed at a multiplicity of infection (MOI) of 10. Bacteria were centrifuged onto the cells for 5 min at 500×g and infection proceeded for 25 min at 37 °C in an atmosphere of 5% CO₂ and 90% humidity. Afterwards, infected cells were washed thrice with PBS and incubated for 1 h with cell culture medium containing 100 μg × ml[−1] gentamicin (Applichem) to kill non-phagocytosed bacteria. Afterwards, the cell culture medium was replaced by medium containing 10 μg × ml[−1] gentamicin until the end of the experiment. If cefotaxime was used during infection experiments, cells were washed with PBS before addition of cell culture medium containing 200 μg × ml[−1] freshly dissolved cefotaxime.

**Host cell infection for microscopy**. RAW264.7 or LAMP1-GFP RAW264.7 macrophages were seeded in surface-treated 24-well plates on glass cover slips to reach 80% confluency (ca. $3.6 × 10^5$ cells per well) on the day of infection. Cells were infected with STM strains as described above at MOI 50 for 8 or 24 h. Afterwards, the cells were washed thrice with PBS and fixed with 3% PFA in PBS for 15 min at RT. After that, cells were directly prepared for subsequent immunostaining.

**Immunostaining and imaging**. Immunostaining of intracellular STM was performed as described before[58]. After fixation of cells with 3% PFA in PBS cells were washed thrice with PBS and directly incubated in blocking solution containing 2% goat serum, 2% bovine serum albumin, and 0.1% saponin in PBS for 30 min at RT. STM was stained with anti-Salmonella O-antigen group B factors 1, 4, 5, 12 (BD Difco, diluted 1:500 in blocking solution) for 1 h at RT. Subsequently, cells were washed thrice in drops of PBS following incubation with secondary antibody Cy5-coupled goat anti-rabbit IgG (Jackson Immuno Research, 1:1000 in blocking solution) for 1 h at RT in the dark. Afterwards, cells were washed thrice, mounted with Fluoroprep (bioMérieux) and sealed with Entellan (Merck). Fluorescence imaging was performed using the confocal laser-scanning microscope Leica SP5. Image acquisition was performed using the 100× objective (HCX PL APO CS 100×; numerical aperture: 1.4–0.7) and the polychroic mirror TD 488/543/633 for the three channels GFP, DsRed, and Cy5 (Leica, Wetzlar, Germany). For setting adjustment, image acquisition and image processing the software LAS AF (Leica, Wetzlar, Germany) was used.

**Flow cytometry analysis**. FC of liberated STM from host cells was performed as described before[40]. Briefly, FC was performed on an Attune NxT instrument (Thermo Fischer Scientific) at a flow rate of 25 μl × min[−1]. At least 10,000 bacteria were gated by virtue of the constitutive/induced DsRed fluorescence. Per gated STM cells, the intensity of the sfGFP fluorescence was determined and x-medians for the sfGFP intensities were calculated.

For the measurement of liberated bacteria (replicating and non-replicating) from host cells, infected cells were lysed at the indicated time points by 0.5% Triton X-100 in PBS for 10 min at RT with shaking to release the intracellular bacterial population. The lysate was transferred to a test tube and after pelleting of host cell debris by centrifugation for 5 min at 500×g, bacteria were recovered from supernatant. Bacteria were further centrifuged for 5 min at 20,000×g and fixed in 3% PFA in PBS for 15 min at RT. After fixation and a further centrifugation step, fixed bacteria were resuspended in 250 μl 100 mM NH₄Cl in PBS for quenching of residual-free aldehydes. After that, samples were directly subjected to FC.

For detection of regrowth of cefotaxime-treated intracellular non-replicating STM, the whole liberated bacterial population was reinoculated into 3 ml fresh LB medium after the bacterial supernatant was transferred to a clean test tube (Supplementary Fig. S7), and incubated at 37 °C using a roller drum at 60 rpm with aeration. At time point 0, 2, 4, and 6 h post re-inoculation samples were taken, diluted with PBS and fixed with 3% PFA in PBS for 15 min at RT. Then, PFA was removed by centrifugation for 5 min at 20,000×g and the pellet was resuspended in 250 μl of 100 mM NH₄Cl in PBS for quenching of residual-free aldehydes. After

that, samples were directly subjected to FC and amounts of DsRed-positive events (non-replicating STM) was measured and depicted by events per μl.

To measure the minimal detectable threshold of bacterial events at the cytometer, various mixed ratios of constitutive DsRed and sfGFP fluorescent STM were prepared. For that, the optical density ($OD_{600}$) of overnight cultures of STM [p5204] and STM [pWRG167] was determined following dilution of the strains to an $OD_{600}$ of 1 (predicted amounts of bacteria: $1.1 \times 10^9$ bacteria $\times$ ml$^{-1}$). Afterwards, serial dilutions were prepared and mixed ratios of red and green fluorescent STM were prepared in PBS. Either a constant high amount of green fluorescent STM (app. 2,000 events $\times$ μl$^{-1}$) mixed with an equal, 10-fold, 100-fold, 1,000-fold, or 10,000-fold reduced amount of red fluorescent STM or a constant high amount of red fluorescent STM (app. 4,000 events $\times$ μl$^{-1}$) mixed with an equal, 10-fold, 100-fold, 1,000-fold, or 10,000-fold reduced amount of green fluorescent STM was measured by FC. The relative amount of green and red fluorescent STM (events $\times$ μl$^{-1}$) measured, when the equal amount of green and red fluorescent STM was present in the sample was set to 100%. As controls, only green, red or no fluorescent STM were measured.

The gating strategy used in FC analyses of various reporter constructs is displayed in Supplementary Fig. S8.

**Statistics and reproducibility**. All values in this study are given as means ± standard deviations of mean. Statistical analysis was accomplished by SigmaPlot 13 by One-Way ANOVA and significance levels are indicated in figures and legends. $p < 0.05$ was considered as statistically significant.

**Reporting summary**. Further information on research design is available in the Nature Research Reporting Summary linked to this article.

## Data availability
The datasets generated and analyzed in this study are available from the corresponding authors upon reasonable request. Source data underlying Figs. 2–6 are available at the Zenodo repository under https://doi.org/10.5281/zenodo.4592976.

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

## Acknowledgements

This work was supported by the Deutsche Forschungsgemeinschaft though grants in SFB 944, projects P4 and P15. We kindly acknowledge intramural funding by profile line P2: Integrated Science of the University Osnabrück. We like to thank the Hans-Mühlenhoff-Stiftung for support of K.O. We thank Tatjana Reuter and Jennifer Röder for isolation of human macrophages.

## Author contributions

M.S. and M.H. conceived and designed the study. M.S. and M.H. developed bacterial strains and reporter constructs. M.S. und K.O. performed experimental work and analysed data. M.S. and M.H. coordinated the project and wrote the manuscript with input of all authors.

## Funding

## Competing interests

The authors declare no competing interests.
