## [Peer Review File · Communications Biology]

Reviewers' Comments:

Reviewer #1:

Remarks to the Author:

see the attached document

Reviewer #2:

Remarks to the Author:

The primary aims of this manuscript were to 1) determine the proportion of Salmonella that form persister cells following infection of RAW 264.7 and human macrophages, 2) determine their ability to respond to stressors during infection of macrophages, and 3) determine whether persister cells are metabolically active. The authors provide evidence that a substantial proportion of intracellular Salmonella form persister cells and a fraction of those persister cells remain metabolically active and respond to stressors within the phagosome. Additionally, they provide evidence that the proportion of persister cells that are metabolically active changes over time. The authors employed a novel dual fluorescence reporter strategy that enabled them to study these phenomena at the single cell level using a combination of fluorescent microscopy and flow cytometry.

The findings presented in this manuscript provide key insights into the phenotypic diversity of Salmonella residing within macrophages. Additionally, the dual fluorescence reporter strategy employed in this manuscript will be highly adaptable to other members of the field to determine the responses of Salmonella to environmental stressors it encounters throughout the course of infection as well as the ability to track the diverse phenotypes that exist within populations of Salmonella.

There are a few key points that were unclear from the submitted manuscript.

1. There was no mention of Salmonella that are killed by macrophages. A large proportion of Salmonella are killed by both RAW 264.7 and human-derived macrophages. Is there any evidence from either the flow cytometry or microscopic data of Salmonella that succumbed to host defenses such as the respiratory burst?
2. Presumably, there were substantially smaller numbers of viable *dksA* and *SPI2* mutants following infection of macrophages. How did the authors account for these differences and how does that impact the interpretation of the flow cytometry data presented in Fig. 3?
3. Of the promoters fused to *sfgfp*, *msrA* and *trxA* are presumably activated in response to ROS produced by macrophages. The impact of inhibiting ROS production in the RAW 264.7 and human macrophages on *msrA* and *trxA* expression, as well as, monitoring the proportion of replicating and nonreplicating Salmonella would have strengthened this manuscript.
4. A rationale for the selection of the *msrA*, *trxA*, and *htrA* promoters was not provided in the introduction of this manuscript.

Reviewer #3:

Remarks to the Author:

This manuscript by Schulte et al explores persister intracellular Salmonella cells in host mononuclear cells. Replicating and nonreplicating bacteria are monitored by following the expression of fluorescent proteins from a constitutive promoter, arabinose and tetracycline inducible promoters and three different promoters of the *msrA*, *trxA* and *htrA* Salmonella genes. The investigators report that nonreplicating Salmonella express to some extent fluorescent proteins from the stress inducible Salmonella promoters. Expression is however diminished compared to replicating controls. The diminished stress response does not seem to be due to a failure to produce the reporter protein. Similarly, nonreplicating bacteria induced in response to cefotaxime treatment also display decreased stress responses. Murine macrophage cells and human

monocytes induce similar persistence in intracellular Salmonella. The studies, which are well designed and clearly presented, provide novel insights into persistent bacteria. I only have a few minor suggestions. Specifically,

Throughout the manuscript, expression of fluorescent reporters is equated as metabolic state or activity. This is misleading, as metabolically active cells may not express protein. Alternatively, nonreplicating cells with low capacity of protein expression may be capable of expressing several metabolic pathways. I recommend the authors are more explicit in their claims and substitute metabolic activity, metabolism, etc for what is being measured in their assays: biosynthesis of a reporter protein.

The manuscripts often used perceive or perception when referring to sensing. As perception has connotations of consciousness, this term should be replaced for sensing throughout the manuscript.

The investigators cite work by Stapels et al regarding the ability of persisters of Salmonella to translocate SPI2 effector proteins. Investigations have also shown that Salmonella can survive as a persister population in the absence of SPI2 function (e.g., PMID: 16129704). This work should be cited.

sfGFP protein was expressed under control of the stress response genes *msrA*, *trxA* or *htrA*. The reader could benefit from the reasoning behind the selection of these promoters. Citations describing roles for these promoters in the stress response of intracellular Salmonella should be included. The authors may want to consider that the gene products encoded by these genes play important roles in antioxidant defense. Interestingly, *TrxA* is a posttranslational regulator of *SsrB*.

Mutants of *ssaV* and *dksA* are used to select for increase frequency of persister cells. Work supporting roles for *ssaV* and *dksA* in the intracellular replication of Salmonella should be cited (e.g., PMID: 9786193 and PMID: 29930310).

Describe the pEM7 promoter.

Spell o/n cultures in line 161

Sentences spanning lines 168-174 are not clear. Please, rephrase.

In main text, describe antibiotic class and mode of action of cefotaxime.

Line 290 and elsewhere, the authors refer to primary human macrophages, but they mean human monocytes. From the vague description provided in the Materials and Methods, it does not seem that the mononuclear phagocytes isolated from peripheral blood were differentiated. This section in Materials and Methods should be described in more detail. For example, it is not clear how the cells were handled after cryopreservation. The infection model of the human monocytes could also benefit from a more thorough description.

Author's response to reviewer's comments

We thank the three reviewers for the critical assessment of our manuscript, the constructive criticism and the useful hints to improve the presentation of our data. Please find our point-by-point response to each comment in blue font. We have also modified Figures to integrated additional data and provide supplementary information for the reviewers.

Reviewers' comments

Reviewer #1 (Remarks to the Author):

Brief summary:

In the work submitted to *Nature Communications Biology*, Shulte *et al.* investigate the responsiveness (gene promoter activity and fluorescent protein production) of intracellular *Salmonella* Typhimurium **non-replicating cells** to stress. Persister cells are a subpopulation of intracellular salmonellae characterized by their non-dividing state. They are sometimes resistant to typically lethal concentrations of antibiotics. Since the non-replicating status is reversible, the persister cells have the potential to resume growth and produce recurrent infections. Specifically, the authors evaluated whether individual non-replicating *Salmonella* Typhimurium (NR-STm) cells were able to detect and respond to stressors. They used dual reporter plasmids for the detection of the NR-STm population, as well as for measuring the stress response and metabolic activity at a single cell level. They employed different stress response related genes as reporters of responsiveness at a single-cell level and observed that the NR population has a lower response to stress compared to the total intracellular population (replicating + non-replicating). While different genes were used to test the ability to respond to stress, there was never a robust attempt to determine what it is the signal that leads the bacteria to decide entering to a state of lesser response, or if there is a particular stressor that promotes the switch. They observed that the low response to stress in the NR-population was not different in SPI-2 mutants, thus they concluded that switching the phenotype to a persister status gives a benefit to the bacteria, regardless of the secretion of effectors. The most exciting thing as a reader was that they measure a change in metabolic activity in the NR population. This finding was limited to 10 h.p.i. after, which metabolic activity begins to decline. This is interesting since they showed that the lower response to stress is not a consequence of a lower metabolic activity.

Major Concerns

Besides the finding that the NR-population decreases its metabolic activity within the first 8-10 h p.i., there is not a great degree of novelty in this study. Most of the results were confirmatory of the works already published (*Helaine et al 2010; Helaine et al 2014; Staples et al 2018; Schulte et al 2020*). Although this study does a solid job characterizing these populations of bacteria, the approach and findings are largely observational and descriptive and as a reader we are left with little to no sense of mechanism. It has been ten years since Dr. Helaine and colleagues unearthed this phenomenon using methodologies almost identical to here. It would seem reasonable that this study would somehow go beyond the previous studies to establish biological relevance and mechanism. Furthermore, the authors currently have a very similar paper upload to BioRxiv, which describes the approach and method used in this current study.

Since the current work is strictly observational and lacks mechanism we feel that in its current form the article is not adequate for publication in this high impact journal.

Response: The pioneering work of Helaine and coworkers provided novel insight into persister formation by intracellular *Salmonella*. The techniques introduced by their work is

also basis for the methods that were used and refined in our study. The prior work also raised a number of questions, some of which we addressed in our present manuscript. The question of the level of exposure of intracellular persisters and their ability to respond to host cell-imposed stress has not been addressed before and we developed methods to investigate this important issue.

The publication Schulte et al. 2020 PMID: 33222378 that appeared in Cell Microbiol is the in-depth evaluation of the dual-fluorescence reporters. This evaluation was prerequisite for the application to the specific questions addressed in the present manuscript.

We do not agree that this report is observational and descriptive. We conclude on mechanisms resulting in reduced stress exposure of intracellular persister and summarize our models in Fig. 7. The finding of continuing stress sensing and response is highly relevant for the understanding of persister physiology and will stimulate further studies on persisters of other pathogens, as well as new forms of therapeutic interference.

The authors analyze the responsiveness of different STm populations by measuring the synthesis of GFP under the control of a stress-induced promoter of the gene of interest (e.g. *PmsrA*). However, it is well known that under stress conditions there is a global shutdown of protein synthesis in the cell in order to conserve energy and cellular resources. Therefore, we wonder if the lower stress-response observed in NR STM vs R-STM is not due to a reduced protein synthesis, instead of a lower stress response itself. To eliminate this controversy, a good control could be the normalization of the levels of *PmsrA*-GFP with the synthesis of some housekeeping protein, instead of comparing it between both populations.

Response: Please appreciate that we use the external induction of the *tetA* promoter exactly to address this question. The data shown in Fig. 4 demonstrate that protein biosynthesis can be induced by an external stimulus. From these observations we conclude that reduced synthesis of the stress reporters is not consequence of global shutdown of protein biosynthesis, but rather an indicator of reduced stress exposure. This is a key message of our manuscript.

Minor concerns:

- Line 144 and Fig S 1D. The main text refers that NR-STM were detected at 16, 24 and 48 h p.i. while the figure legend on the Fig. S 1D reads as 8, 16 and 24 h p.i. What is the correct time point? The figure is also showing different counts for each time point assayed. Can you draw any conclusion from the time inside the cell and the proportion of NR-population?

Response: The text has been corrected to 8 h, 16 h and 24 h p.i. to match the data displayed in Fig. S1.

As this analysis was used to establish experimental conditions for the subsequent analyses, different counts were obtained. Based on these data we defined which number of cells had to be analysed to obtain comparable numbers of intracellular NR STM.

- Line 150. As a reader it is hard to follow the rationale of using *AssaV* and *AdksA* at this point. These genotypes were not properly introduced in this assay.

Response: We added a section describing the role of *ssaV*, *sifA*, *sseF* and *dksA*, and the known phenotypes of these mutant strains (lines. 154-160). The introduction also refers to the role of DksA in repair of ROS-mediated damages.

- Line 183. (Related to Fig 3A). In the context of this assay did you test other genes than *msrA* in the SPI2-mutants background? If yes, have you seen the same trend?

Response: We performed further experiment with additional stress reporters *htrA* and *trxA*. The data are included in revised Fig. S3HI and indicate comparable effects.

- Line 184. Would it be possible to have a statistical comparison between NR-WT and NR- $\Delta dskA$? It appears that the stress response of the $\Delta dskA$ NR-population is lower than the *Wild-type* NR-population.

Response: Statistics were added for NR and EP values of mutant strains compared to WT strain in Fig. 3. Indeed, the stress response of $\Delta dskA$ NR, as well as of $\Delta ssaV$ and $\Delta sifA$ NR was lower than WT NR.

- Related to Fig. 6. It would be interesting to see the proportion of NR vs R-population in human macrophages. Based on the results obtained with the antibiotic treatment, where the presence of a stressor increases the percentage of NR-STm, we can assume that in human macrophages the proportion will be higher than in RAW cells.

Response: In human macrophages with M1 polarization, we observed extreme reduction of the R population. All detected STM were NR, thus persisters. This is stated in the legend, line 841-842. This finding is in line with high antimicrobial activity of human M1 macrophages and the inability of STM to proliferate in these host cells.

- Line 259. After using cefotaxime as a stressor, the proportion of NR bacteria increased about 41%, while the stress response in NR and the entire population decreased. Have you tried testing other intracellular antibiotics, for example Ciprofloxacin, Levofloxacin? (the latter are used as clinical treatment).

Response: We also performed analyses using Ciprofloxacin. The results are shown below. Ciprofloxacin showed similar effects on stress response reporters. However, Ciprofloxacin resulted in less reduction of R population of intracellular STM and we continued using Cefotaxim. Cefotaxim has been used in a number of related studies.

In vitro killing of STM using cefotaxim and ciprofloxacin

Overnight culture of STM was subcultured (1/31) in LB
 Incubated at 37 degree celsius for 2 h
 Addition of antibiotics 2 h post subculturing and incubation for 2 h

Intracellular stress induction of STM after cefotaxim / ciprofloxacin treatment

Infection of RAW264,7 macrophages for 24 h
 Antibiotics were added 10 h p.i.
 At 24 h p.i. host cells were lysed and samples were prepared for FC as mentioned before
 Density plots for untreated and cefotaxim-treated STM was taken from figure S6

- In the results section, it would be helpful for the authors to provide a justification and formulate a specific hypothesis in each paragraph before describing their experiments in more detail. For example, during the reading process we had to turn to previous publications first in order to understand the connection between the different mutants used here.

Response: Thank you for this suggestion. There is always the balance between detailed description and length restriction. We now added brief introductions on the hypothesis. In the revised version, the mutant strains used in the study are better described, also in response to

- Line 166. It could be nice to have a brief explanation of those genes, the names, their role in the stress response, and maybe why did you choose them. E.g. include a scheme of the plasmid plus the name of the genes of interest, as it is drawn in Schulte et al 2020. In that publication

you visually explain the type of stress that you are measuring and the name of the gene that you are assessing.

Response: we modified Fig. 1 accordingly and also provide more details on the selection of the reporters. See also response to reviewer 3, point 4. We also give reference to our publication that provide full details of the construction and in-depth characterization of the reporters.

Reviewer #2 (Remarks to the Author):

The primary aims of this manuscript were to 1) determine at the proportion of Salmonella that form persister cells following infection of RAW 264.7 and human macrophages, 2) determine their ability to respond to stressors during infection of macrophages, and 3) determine whether persister cells are metabolically active. The authors provide evidence that a substantial proportion of intracellular Salmonella form persister cells and a fraction of those persister cells remain metabolically active and respond to stressors within the phagosome. Additionally, they provide evidence that proportion of persister cells that are metabolically active changes over time. The authors employed a novel dual fluorescence reporter strategy that enabled them to study these phenomena at the single cell level using a combination of fluorescent microscopy and flow cytometry. The findings presented in this manuscript provide key insights into the phenotypic diversity of Salmonella residing within macrophages. Additionally, the dual fluorescence reporter strategy employed in this manuscript will be highly adaptable to other members of the field to determine the responses of Salmonella to environmental stressors it encounters throughout the course of infection as well as the ability to track the diverse phenotypes that exist within populations of Salmonella.

There are a few key points that were unclear from the submitted manuscript.

1. There was no mention of Salmonella that are killed by macrophages. A large proportion of Salmonella are killed by both RAW 264.7 and human-derived macrophages. Is there any evidence from either the flow cytometry or microscopic data of Salmonella that succumbed to host defenses such as the respiratory burst?

Response: This is a good point, but it is difficult to address experimentally. We consider that ability to form colonies is the best proof of viability. To do this on the level of single intracellular bacteria with prior interrogation of stress reporter activation, FACS for sorting of individual bacteria and test of colony formation is required. Such analyses demand specific instrumentation (FACS for BL2 applications) and justify separate investigations. We are not aware of in situ analyses such as fluorescent probes that would reliably distinguish dead bacteria from persisters. This may explain our very limited use 'killed' and 'dead'.

2. Presumably, there were substantially smaller numbers of viable *dksA* and SPI2 mutants following infection of macrophages. How did the authors account for these differences and how does that impact the interpretation of the flow cytometry data presented in Fig. 3?

Response: Indeed, the absolute number of the *dksA* and SPI2 mutant strains is reduced. Since we analysed the same number of bacteria for all conditions compared, the data allow direct

comparison. For all analyses, at least 10,000 bacterial cells were analysed. The number of infected host cells was sufficiently high to yield the required number of intracellular STM, for WT as well as for attenuated strains.

3. Of the promoters fused to *sfgfp*, *msrA* and *trxA* are presumably activated in response to ROS produced by macrophages. The impact of inhibiting ROS production in the RAW 264.7 and human macrophages on *msrA* and *trxA* expression, as well as, monitoring the proportion of replicating and nonreplicating Salmonella would have strengthened this manuscript.

Response: We performed the suggested experiment using a strain harboring the *msrA* reporter, and treatment by DPI for inhibition of ROS production. The data show that *msrA* induction is highly reduced in the entire population if DPI is applied, but no change was observed for the NR population. (Fig. S3 G)

4. A rationale for the selection of the *msrA*, *trxA*, and *htrA* promoters was not provided in the introduction of this manuscript.

Response: We added a section in the introduction describing the stress response function of MsrA, TrxA and HtrA (lines 79-87). Also, Figure 1 was modified to provide a link between the promoters used and the stress response functions. The rationale for selection of these promoters is given in the first paragraph of the results section, with reference to work that comprehensively characterized the stress reporters (lines 121-123).

Reviewer #3 (Remarks to the Author):

This manuscript by Schulte et al explores persister intracellular Salmonella cells in host mononuclear cells. Replicating and nonreplicating bacteria are monitored by following the expression of fluorescent proteins from a constitutive promoter, arabinose and tetracycline inducible promoters and three different promoters of the *msrA*, *trxA* and *htrA* Salmonella genes. The investigators report that nonreplicating Salmonella express to some extent fluorescent proteins from the stress inducible Salmonella promoters. Expression is however diminished compared to replicating controls. The diminished stress response does not seem to be due to a failure to produce the reporter protein. Similarly, nonreplicating bacteria induced in response to cefotaxime treatment also display decrease stress responses. Murine macrophage cells and human monocytes induce similar persistence in intracellular Salmonella. The studies, which are well designed and clearly presented, provide novel insights into persistent bacteria.

I only have a few minor suggestions.

1. Specifically, throughout the manuscript, expression of fluorescent reporters is equated as metabolic state or activity. This is misleading, as metabolically active cells may not express protein. Alternatively, nonreplicating cells with low capacity of protein expression may be capable of expressing several metabolic pathways. I recommend the authors are more explicit

in their claims and substitute metabolic activity, metabolism, etc for what is being measured in their assays: biosynthesis of a reporter protein.

Response: Our analyses test the ability to synthesize fluorescent proteins (FP) and use this property as proxy of biosynthetic capacity and metabolic activity. This is explained in lines 223-224. By using an external experimental stimulus to induce FP synthesis, this biosynthesis is uncoupled other responses of the cell, yet dependent on the energy charge and overall physiological state of intracellular STM.

2. The manuscripts often used perceive or perception when referring to sensing. As perception has connotations of consciousness, this term should be replaced for sensing throughout the manuscript.

Response: perceptual, perceive and perceiving was replaced by sensing throughout the manuscript.

3. The investigators cite work by Stapels et al regarding the ability of persisters of Salmonella to translocate SPI2 effector proteins. Investigations have also shown that Salmonella can survive as a persister population in the absence of SPI2 function (e.g., PMID: 16129704). This work should be cited.

Response: We addressed this point in the discussion, cited the publication and confronted the finding with observed activity of SPI2-T3SS in persisters of STM. Lines 334-340.

4. sfGFP protein was expressed under control of the stress response genes *msrA*, *trxA* or *htrA*. The reader could benefit from the reasoning behind the selection of these promoters. Citations describing roles for these promoters in the stress response of intracellular Salmonella should be included. The authors may want to consider that the gene products encoded by these genes play important roles in antioxidant defense. Interestingly, TrxA is a posttranslational regulator of SsrB.

Response: We modified Fig. 1 to introduce the function of *msrA*, *trxA* or *htrA*. We also added sections in introduction and results to give further explanation and references. See also response to reviewer 2, point 4.

5. Mutants of *ssaV* and *dksA* are used to select for increase frequency of persister cells. Work supporting roles for *ssaV* and *dksA* in the intracellular replication of Salmonella should be cited (e.g., PMID: 9786193 and PMID: 29930310).

Response: Citations to these studies are added at appropriate positions

6. Describe the pEM7 promoter

Response: Description of this promoter was added to the results (line 116-117) and the legend of Fig. 1

7. Spell o/n cultures in line 161

Response: Abbreviation o/n for overnight was introduced on line 138.

8. Sentences spanning lines 168-174 are not clear. Please, rephrase.

Response: This section was rephrased for clarity

9. In main text, describe antibiotic class and mode of action of cefotaxime.

Response: mode of action introduced in lines 282-293.

10. Line 290 and elsewhere, the authors refer to primary human macrophages, but they mean human monocytes. From the vague description provided in the Materials and Methods, it does not seem that the mononuclear phagocytes isolated from peripheral blood were differentiated. This section in Materials and Methods should be described in more detail. For example, it is not clear how the cells were handled after cryopreservation. The infection model of the human monocytes could also benefit from a more thorough description.

Response: Thank you for pointing this out. Indeed, after isolation, the monocytes were differentiated to macrophages for use in infection experiment. We added the missing part of the procedure (lines 497-502).

Reviewers' Comments:

Reviewer #1:

Remarks to the Author:

After reviewing in detail, the rebuttal from Shulte et al., I consider that the findings provided by this research (about the bacterial ability to sense stressors within infected cells) are pertinent to understanding the physiology of persisters. Thus, this information will lead to future knowledge and new antimicrobial molecules capable of attacking intracellular subpopulations of *Salmonellae*. However, I still consider the work to be primarily observational and descriptive, although this does not make it less important or less interesting. The authors present several well-conducted and controlled experiments in which they observe that those bacteria in the persister stage are metabolically active and can respond to stressful situations, however they do not elucidate the mechanism itself. Nevertheless, I consider that the work of moderate impact and could be published in your journal.

Regarding the revised version, the authors have considered and included all the minor suggestions made, including modifications to the main text, fig legends, statistics, as well as the addition of some experiments (Fig. S 3G), and Fig for the reviewer.

Reviewer #2:

Remarks to the Author:

Thank you for the thoughtful responses to the comments raised during the initial review.

Reviewer #3:

Remarks to the Author:

The authors have addressed all my queries.